# The impact of the SARS-CoV-2 pandemic on the demographic, clinical and social profiles of patients admitted to the Pneumology Department for a COPD exacerbation

Alberto Fernández Villar[1,2], Rafael Golpe Gómez [3], Almudena González Montaos [1,2]*, Sara Fernández García[2], Luis Pazos Area[1,2], Ana Priegue Carrera[2], Alberto Ruano Raviña[4,5], Cristina Represas Represas[1,2]

1 Pulmonology Department, Álvaro Cunqueiro University Hospital, Vigo, Spain, 2 Institute of Health Research Galicia Sur (IISGS), Neumo I + i. Research Group, Pontevedra, Spain, 3 Pulmonology Department, Lucus Augusti University Hospital, Lugo, Spain, 4 Department of Preventive Medicine and Public Health, University of Santiago de Compostela, Santiago, Spain, 5 Consortium for Biomedical Research in Epidemiology and Public Health (CIBER en Epidemiología y Salud Pública—CIBERESP), Barcelona, Spain

* almudena.gonzalez.montaos@sergas.es

## Abstract

### Introduction

Although a reduction in admissions for pathologies other than SARS-CoV-2 has been reported during the pandemic, there are hardly any specific studies in relation to COPD. The objective of this study was to analyse differences in the profile of those admitted for AEPOC and their prognosis during this period.

### Methods

Prospective study (SocioEPOC validation cohort) conducted in two hospitals. Demographic, clinical and social characteristics were compared among patients admitted for an AECOPD before and after the declaration of the COVID-19 healthcare emergency. Mortality and the need for hospital care in the following 3 months were analysed.

### Results

340 patients (76.6% male, 72 years, FEV1 43.5%) were included, 174 in the post-pandemic phase. During pandemic, especially before population-level vaccination, admissions for AECOPD were in patients with more severe disease and with a higher level of eosinophils. No differences were found in social profile, except they had more informal caregivers. The mortality rate at 90 days was the same (9%), although those admitted during the pandemic came for more hospital visits in the following 3 months (53.8% vs. 42%; $p = 0.003$), with the pandemic phase being an independent predictor of this possibility (OR = 1.6.; 95% IC = 1.1–2.6).

**Data Availability Statement:** All relevant data are within the paper and its Supporting Information files.

**Funding:** This study has been funded by Instituto de Salud Carlos III through the project PI8/01317 (Cofunded by European Regional Development Fund "A way to make Europe") and the unconditional collaboration of the company Menarini. The funders had no role in study design, data collection and analysis, decision to publish, or preparation of the manuscript.

**Competing interests:** None of the authors declare conflicts of interest for the contents of this manuscript. AFV has received honoraria in the last 3 years for lecturing, scientific consulting, participating in clinical studies, or writing publications for (alphabetical order): AstraZeneca, Boehringer Ingelheim, Chiesi, GlaxoSmithKline, Grifols, and Menarini. CRR has received honoraria in the past 3 years for lecturing, scientific consulting, clinical trial participation, or publication writing for (alphabetical order): AstraZeneca, Boehringer Ingelheim, Chiesi, Faes farma, and GlaxoSmithKline. This does not alter our adherence to PLOS ONE policies on sharing data and materials.

## Conclusions

In the first few months of the pandemic, the clinical profile of patients hospitalised for an AECOPD differed from that both prior to this period and during the latter months of the pandemic, with minimal changes at the social level. Although the mortality rate were similar, unscheduled hospital visits increased during the COVID-19 pandemic.

## Introduction

Chronic obstructive pulmonary disease (COPD) causes significant morbidity and mortality and consumes a lot of health resources, mainly because acute exacerbations of COPD (AECOPD) requiring hospitalisation [1,2]. Since the declaration of the COVID-19 pandemic, multiple publications have reported a 40–60% decrease in cases of severe AECOPD not caused by the SARS-CoV-2. This has been attributed to a decrease in intercurrent infections because of the respiratory protection measures and social restrictions imposed in many countries [3–11]. The reduction in industrial and urban pollution, greater therapeutic adherence or a decrease in attendance to health centers due to the overload and fear of infection may be other contributing factors to this change [3,11,12].

The impact of the COVID-19 pandemic on patients with advanced chronic pathologies was notable, affecting both the clinical, psychological and social spheres. This included the care and social support received during this time, which impacted patient quality of life, especially among those in more disadvantaged social situations [13–15]. Studies specifically focussing on COPD during pandemic are scarce and involved qualitative research [14,15]. Whether the demographic, clinical and social profiles of patients with a severe AECOPD during the COVID-19 pandemic differed from those previously admitted is a matter that has been little studied.

The findings from the few studies that did analyse prognostic aspects of COPD such as hospital mortality rates did not coincide [6–9] and failed to study important factors such as readmissions or deaths in the months following hospitalisation. In addition, whether changes occurred in any of these factors after the general vaccination of the population against SARS-CoV-2 remains unknown. Thus, we conducted this work to compare the demographic, clinical, prognostic and social characteristics of a cohort of patients with severe an AECOPD seen both in the months before and after the declaration of the COVID-19 pandemic and to assess whether vaccination of the population against SARS-CoV-2 influenced any of these factors.

## Materials and methods

### Design and setting

This was a prospective study that included patients between November 2019 and June 2022 to create the SocioCOPD validation cohort (the referral cohort was created in 2017) [16,17]. The aim of this work was to analyse the social and clinical profiles of patients with a severe AECOPD and to study the influence of these characteristics on their prognoses. Patients with an index admission (first admission within the study period) with the primary diagnosis of a severe AECOPD at one of two Pneumology Departments were consecutively included. These departments were located in two different public hospitals in a region of north-western Spain with 375,000 and 205,000 populations.

Patients who did not agree to participate or with alternative diagnoses to COPD or severe AECOPD during admission or follow-up were excluded. From March 2020, any patients with

a SARS-CoV-2 infection confirmed by microbiological techniques (which were performed on every patient) were also excluded from this research. During the first few days of hospitalisation, we interviewed the patients and their informal caregivers and completed the information we required by reviewing their electronic medical records. This study was approved by the Galician Research and Ethics Committee and all the patients signed their informed consent to participation.

## Information collection and definition of variables

We registered age, sex, body mass index, influenza and pneumococcal vaccination status from the previous year, previous hospitalisations for AECOPD or other reasons, positive sputum cultures from the year prior and their number, disease impact, degree of dyspnoea prior to an AECOPD using the COPD Assessment Test (CAT) questionnaire and modified Medical Research Council (mMRC) scale, FEV1 value in the last spirometry performed and need for home oxygen therapy or non-invasive ventilation before hospitalisation. The coexistence and number of comorbidities included in the Charlson index, as well as any previous diagnoses of anxiety or depression were also recorded. Peripheral blood eosinophil levels, hospital length of stay, mortality rate, respiratory failure at discharge and the recommended pharmacological treatments were also collected.

From the social perspective, place of residence and cohabitation status, including whether they lived or slept alone, social relationships (family, neighbours and friends), need for social worker visits, home ownership, monthly economic income, availability of own transport and need for an informal or paid caregiver was included. Dependency for basic activities was evaluated using 6 of the 10 Barthel index variables (feeding, dressing, bathing, toilet use, going up/down stairs and chair transferring) [18], as well as the 8 instrumental variables included in the Lawton and Brody index (house cleaning, food preparation, laundry, telephone use, shopping activity, managing finances, taking medication and using public transport) [19].

Emergency Department visits, hospital readmissions and death were followed-up at 30 and 90 days via their electronic medical records and by telephone. The declaration of the COVID-19 pandemic allowed us to spit the cohort into two groups. Patients included up until 14 March 2020 (declaration of health emergency in Spain) were considered in the group from before the pandemic while those registered after this date formed the pandemic group. Patients from the latter group were further divided into the pre-SARS-CoV-2 vaccination group if they were hospitalised between 14 March 2020 and 30 June 2021 and post-vaccination group. By this time, 85% of the populations in the health areas of the participating hospitals had received at least one dose of the vaccine while more than 70% aged over 50 years had received the full regimen [20].

## Statistical analysis

We verified the adjustment of the quantitative variables to normality using the Shapiro–Wilk test and were expressed as median and 25th and 75th percentiles. The qualitative variables were expressed as absolute value and percentages. Numerical variables were compared using Mann–Whitney U tests and qualitative variables using Chi-squared or Fisher exact tests. To assess whether the period of admission (pre-pandemic and pandemic) was an independent factor for readmission at 90 days, we performed a logistic regression analysis in which we included all the variables that presented a $p \leq 0.05$ for this event in the univariate study. We calculated the odds ratios (OR) with their 95% confidence intervals (95% IC) for all the variables. The numerical variables included in the model were dichotomised based on their median values. SPSS software for Windows (version 25; IBM Corp, Armonk, NY, USA) was used for analysis.

## Results

The follow-up of the patients included in the study is described in the flowchart of Fig 1.

340 patients were included in this work, 166 in the pre-pandemic phase and 174 in the ongoing pandemic phase, of which 87 cases were considered within the pre-vaccination period. Table 1 includes the demographic and clinical characteristics and Table 2 shows the social variable data and a comparison of the hospitalised patients in the different periods and sub-periods considered. Only 6 (1.8%) patients lived in communal residences and none had significant cognitive impairments.

Patients hospitalised in the months prior to the pandemic had a higher number of admissions and positive sputum cultures in the previous year and a lower frequency of influenza vaccinations. Differences were also observed in the degree of dyspnoea, coexistence and number of comorbidities and need for home support therapies, which was higher in patients admitted in the initial months of the pandemic compared to the post-vaccination phase or the pre-pandemic stage. In addition eosinophil levels in blood were higher in patients hospitalised during the pandemic, especially in the first few months.

Regarding the social variables during the pandemic, the informal caregiver became more common and fewer patients went out alone, especially in its initial phase. Less dependence for instrumental activities was evidenced during this period, especially for housework like doing laundry or food preparation, but not for telephone use, financial management or taking medication, in which no differences were described (data not shown). There were no differences in the economic situation of the participants or their social relationships.

Table 3 shows the length of hospital stay, Emergency Department visits and hospital readmissions in the short and medium term, as well as the mortality rate during hospitalisation

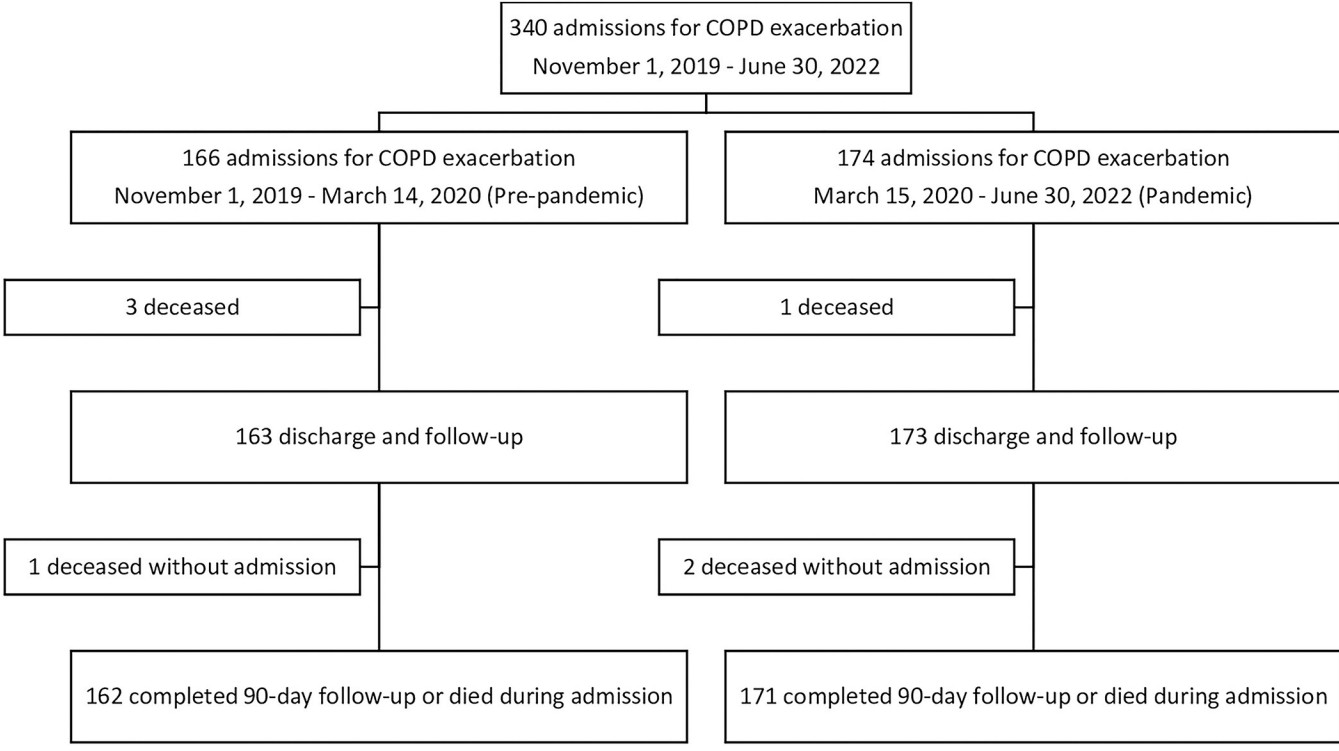

**Fig 1. Patient follow-up flowchart.**

**Table 1. Demographic, clinical and social characteristics of patients admitted for COPD exacerbation before and during the pandemic, and during this period in the pre-vaccination or vaccination phase.**

| Variables | Total (340) | Pre-pandemic (166) (1) | Pandemic (174) (2) | P (1 vs 2) | Pandemic: pre-vaccination phase (87) (3) | Pandemic: post-vaccination phase (87) (4) | P (1 vs 3) | P (1 vs 4) | P (3 vs 4) |
|---|---|---|---|---|---|---|---|---|---|
| Male (sex) (%) | 264 (76.6) | 126 (75.9) | 138 (79.3) | 0.51 | 72 (82.8) | 66 (75.9) | 0.26 | >0.99 | 0.35 |
| Age (years)[a] | 72 (64–77) | 71 (63–77) | 72 (65–79) | 0.07 | 72 (66–79) | 72 (65–79) | 0.12 | 0.15 | 0.85 |
| BMI (kg/m$^2$)[a] | 27.9 (23.4–31.9) | 27.6 (23.5–31.4) | 27.9 (23.1–32) | 0.86 | 28.4 (23.5–32.7) | 27 (22.6–31.6) | 0.59 | 0.67 | 0.38 |
| Smoker/ Ex-smoker | 320 (94.1) | 160 (96.4) | 160 (92) | 0.11 | 83 (95.4) | 77 (88.5) | 0.74 | 0.10 | 0.16 |
| Active smoker (%) | 122 (38.2) | 68 (42.5) | 51 (34) | 0.07 | 26 (31.3) | 28 (36.8) | 0.09 | 0.47 | 0.50 |
| Pack-year[a] | 53 (40–80) | 50 (40–72) | 60 (40–80) | 0.03 | 60 (40–80) | 60 (40–80) | 0.21 | 0.12 | 0.73 |
| ≥1 admission within the previous year for any reason (%) | 157 (46.2) | 84 (50.6) | 73 (42) | 0.12 | 41 (47.1) | 32 (36.8) | 0.69 | 0.04 | 0.21 |
| Number of admissions within the previous year for any reason[a] | 0 (0–1) | 1 (0–1) | 0 (0–1) | 0.09 | 0 (0–1) | 0 (0–1) | 0.62 | 0.02 | 0.12 |
| ≥1 admission within the previous year for COPD exacerbation (%) | 119 (35) | 70 (42) | 49 (28.2) | 0.009 | 29 (33.3) | 20 (23) | 0.17 | 0.002 | 0.17 |
| Number of admissions within the previous year for COPD exacerbation[a] | 0 (0–1) | 0 (0–1) | 0 (0–1) | 0.005 | 0 (0–1) | 0 (0–0) | 0.21 | 0.001 | 0.07 |
| Positive sputum culture within the previous year to admission (%) | 76 (22.4) | 44 (26.5) | 32 (18.5) | 0.09 | 21 (24.4) | 11 (12.6) | 0.76 | 0.01 | 0.05 |
| Number of positive sputum culture within the previous year to admission[a] | 0 (0–0) | 0 (0–0) | 0 (0–0) | 0.11 | 0 (0–0) | 0 (0–0) | 0.86 | 0.01 | 0.04 |
| At-home oxygen therapy (%) | 134 (39.5) | 61 (36.7) | 73 (42.2) | 0.31 | 42 (48.8) | 31 (35.6) | 0.07 | 0.89 | 0.07 |
| At-home non-invasive mechanical ventilation (%) | 32 (9.5) | 19 (11.4) | 13 (7.6) | 0.26 | 11 (12.9) | 2 (2.3) | 0.83 | 0.01 | 0.009 |
| Influenza vaccination (%) | 260 (76.6) | 120 (72.3) | 140 (81) | 0.05 | 72 (83.7) | 68 (78.2) | 0.04 | 0.36 | 0.44 |
| Pneumococcal vaccination (%) | 231 (68.5) | 11 (67.7) | 120 (69.8) | 0.64 | 55 (64.7) | 65 (74.7) | 0.77 | 0.25 | 0.18 |
| Coexistence of some comorbidity (%) | 214 (63.1) | 110 (66.3) | 104 (60.1) | 0.26 | 63 (77.3) | 41 (47.1) | 0.31 | 0.004 | 0.01 |
| Number of comorbidities[a] | 2 (1–3) | 2 (1–3) | 2 (1–3) | 0.78 | 2 (1–3) | 1 (0–2) | 0.37 | 0.18 | 0.03 |
| CAT Score[a] | 23 (17–28) | 23 (18–28) | 20 (16–27) | 0.10 | 20 (16–26) | 22 (16–27) | 0.06 | 0.18 | 0.78 |
| Dyspnea mMRC[a] | 2 (2–3) | 2 (2–3) | 2 (2–3) | 0.87 | 2 (2–3) | 2 (1–3) | 0.06 | 0.04 | 0.001 |
| Dyspnea mMRC 3–4 (%) | 133 (39.8%) | 68 (41.5) | 65 (38.2) | 0.57 | 42 (49.4) | 23 (27.1) | 0.28 | 0.02 | 0.004 |
| FEV$_1$ value (mL) | 1110 (870–1510) | 1090 (825–1480) | 1150 (870–1580) | 0.32 | 1060 (880–1472) | 1260 (810–1630) | 0.75 | 0.19 | 0.48 |
| FEV$_1$ value (% reference)[a] | 43.5 (32–56) | 42 (31–54) | 44 (33–57) | 0.19 | 43 (33–52) | 42 (31–54) | 0.78 | 0.08 | 0.28 |
| Eosinophils at admission (total; cells·µL$^{-1}$)[a] | 100 (37–200) | 100 (15–150) | 110 (70–205) | <0.001 | 120 (90–300) | 120 (60–200) | <0.001 | 0.05 | 0.05 |
| Eosinophils at admission (%)[a] | 1 (0.3–1.9) | 0.9 (0.1–1.5) | 1.2 (0.5–2) | <0.001 | 1.6 (0.9–2.5) | 1 (0.5–1.7) | <0.001 | 0.09 | 0.006 |
| Eosinophils (cells·µL$^{-1}$) range at admission:<br>• ≥ 300<br>• 100–299<br>• < 100 | 51 (15.1)<br>149 (44.1)<br>138 (40.1) | 17 (10.3)<br>66 (40)<br>82 (49.7) | 34 (19.7)<br>83 (48)<br>56 (32.4) | 0.002 | 23 (26.7)<br>38 (44.2)<br>25 (29.1) | 11 (12.6)<br>45 (51.7)<br>31 (35.6) | <0.001 | 0.89 | 0.05 |
| Respiratory failure at discharge (%) | 175 (52.2) | 77 (47.5) | 98 (56.6) | 0.10 | 50 (58.1) | 48 (55.2) | 0.71 | 0.28 | 0.76 |

[a]Shown as median and 25th and 75th percentiles.

BMI: Body Mass Index, CAT: COPD Assessment Test; mMRC: Modified Medical Research Council dyspnea scale, FEV$_1$: forced expiratory volume in the first second.

**Table 2. Social characteristics in patients admitted for COPD exacerbation before and during the pandemic, and during this period in the pre-vaccination or vaccination phase.**

| Variables | Total (340) | Pre-pandemic (166) (1) | Pandemic (174) (2) | P (1 vs 2) | Pandemic: pre-vaccination phase (87) (3) | Pandemic: post-vaccination phase (87) (4) | P (1 vs 3) | P (1 vs 4) | P (3 vs 4) |
|---|---|---|---|---|---|---|---|---|---|
| Secondary/University studies (%) | 59 (17.4) | 31 (18.6) | 28 (16.2) | 0.54 | 11 (12.6) | 17 (20) | 0.42 | 0.72 | 0.58 |
| Rural area of residence (%) | 174 (51.8) | 85 (53.9) | 86 (49.7) | 0.44 | 44 (51.2) | 42 (48.3) | 0.69 | 0.42 | 0.76 |
| Home ownership (%) | 224 (66.9) | 102 (62.6) | 122 (70.9) | 0.24 | 56 (64.4) | 66 (77.6) | 0.35 | 0.11 | 0.27 |
| Monthly income < 800€ (%) | 193 (57.5) | 94 (56.6) | 95 (55.9) | 0.91 | 53 (60.9) | 46 (55.4) | 0.61 | 0.92 | 0.52 |
| Employment situation: pensioner (%) | 315 (92.6) | 194 (92.8) | 161 (92.5) | 0.68 | 79 (90.8) | 82 (94.3) | 0.99 | 0.53 | 0.21 |
| No social relationships or only family (%) | 47 (13.8) | 22 (13.3) | 25 (14.4) | 0.87 | 12 (13.8) | 13 (14.9) | >0.99 | 0.76 | >0.99 |
| Live alone (%) | 52 (15.3) | 24 (14.5) | 28 (16.1) | 0.76 | 13 (14.9) | 15 (17.2) | >0.99 | 0.59 | 0.83 |
| Sleep alone (%) | 120 (35.5) | 53 (32.1) | 67 (38.7) | 0.21 | 35 (40.2) | 32 (37.2) | 0.21 | 0.48 | 0.75 |
| Have a caregiver (%) | 207 (60.9) | 96 (57.8) | 111 (63.8) | 0.26 | 56 (64.4) | 55 (63.2) | 0.34 | 0.42 | 0.99 |
| Informal caregiver (%)[a] | 182 (86.7) | 78 (79.6) | 104 (92.9) | 0.007 | 52 (92.9) | 52 (92.9) | 0.03 | 0.03 | >0.99 |
| Drive (%) | 147 (43.2) | 72 (43.4) | 75 (43.1) | 0.99 | 38 (43.7) | 37 (42.5) | >0.99 | >0.99 | >0.99 |
| Go outside alone (%) | 274 (80.6) | 141 (84.9) | 133 (76.4) | 0.05 | 66 (75.9) | 67 (77) | 0.08 | 0.12 | >0.99 |
| Dependency for basic activities (%) | 98 (28.8) | 47 (28.3) | 51 (29.0) | 0.90 | 27 (31) | 24 (26.7) | 0.66 | >0.99 | 0.73 |
| Number of basic activities with dependency (%) | 0 (0–1) | 0 (0–1) | 0 (0–1) | 0.42 | 0 (0–2) | 0 (0–1) | 0.34 | 0.71 | 0.61 |
| Dependency for instrumental activities (%) | 237 (69.7) | 126 (75.9) | 111 (63.8) | 0.01 | 56 (64.4) | 55 (63.2) | 0.03 | 0.04 | >0.99 |
| Previous use of social services resources (%) | 84 (24.9) | 40 (24.2) | 44 (25.4) | 0.86 | 24 (27.9) | 20 (23) | 0.54 | 0.87 | 0.48 |

[a]Applied to 207 patients who had a caregiver.

and in the follow-up period. A trend towards longer hospital stays during the pandemic was observed, although this did not reach statistical significance. No differences in mortality were described either during hospitalisations or in the subsequent months. Four patients died while hospitalised and three died in an outpatient setting during the 90-day follow-up. 160 of the remaining 333 patients needed emergency care or were readmitted within the first 90 days of follow-up after discharge, with 3 deaths during readmission. A higher frequency of Emergency Department visits and/or readmissions was observed both in the short and medium term for cases of a severe AECOPD during the pandemic.

Table 4 shows the univariate and logistic regression analysis of the main sociodemographic and clinical variables that were predictive of need for emergency care or readmission 90 days after discharge. Only grade 3–4 dyspnoea (OR = 1.8; 95% IC = 1.1–3.3) and hospitalisation during the pandemic period (OR = 1.6; 95% IC = 1.1–2.6) still significantly predicted the need for emergency care or readmission during the 90 days after discharge.

**Table 3. Days of hospital stay, mortality and Emergency Department visits or hospital readmissions in patients admitted for COPD exacerbation before and during the pandemic, and during this period in the pre-vaccination or vaccination phase.**

| Variables | Total (340) | Pre-pandemic (166) (1) | Pandemic (174) (2) | P (1 vs 2) | Pandemic: pre-vaccination phase (87) (3) | Pandemic: post-vaccination phase (87) (4) | P (1 vs 3) | P (1 vs 4) | P (3 vs 4) |
|---|---|---|---|---|---|---|---|---|---|
| Days of hospital stay[a] | 6 (4–9) | 6 (4–8) | 6.5 (4–9) | 0.09 | 6 (4–9) | 7 (4–9) | 0.09 | 0.29 | 0.62 |
| Death during admission (%) | 4 (1.2) | 3 (1.8) | 1 (0.6) | 0.36 | 1 (1,1) | 0 | >0.99 | 0.55 | >0.99 |
| Emergency Department visits with/ without readmission after 30 days (%) | 83 (24.6) | 32 (19.4) | 51 (29.5) | 0.03 | 22 (25.6) | 29 (33.3) | 0.25 | 0.02 | 0.31 |
| Death in the first 30 days after discharge (%) | 6 (1.7) | 3 (1.8) | 3 (1.7) | 0.99 | 0 | 3 (3.4) | 0.92 | 0.29 | 0.24 |
| Emergency Department visits with/ without readmission at 90 days (%) | 160 (48) | 68 (42) | 92 (53.8) | 0.03 | 47 (54.7) | 45 (52.9) | 0.06 | 0.10 | 0.87 |
| Death in the first 90 days after discharge (%) | 22 (6.5) | 9 (5.5) | 13 (7.5) | 0.51 | 5 (5.8) | 8 (9.2) | >0.99 | 0.29 | 0.56 |

[a]Shown as median and 25th and 75th percentiles.

## Discussion

Although the evidence is clear on the significant reduction in hospitalisations for AECOPD during the pandemic, there was very little evidence available about the profile of these patients [3–10]. This study provides novel information on the demographic, clinical and social characteristics of patients admitted for an AECOPD not in relation to SARS-CoV-2, as well as their evolution during the pandemic compared to those admitted in the previous months.

The results showed that patients admitted during the pandemic, especially before mass vaccination, presented more severe disease and higher peripheral blood eosinophil levels. At a social level, they had informal caregivers more frequently and fewer limitations for instrumental activities, although they left their homes less frequently. Lastly, the frequency of hospital visits was higher during this period, even after adjusting for other variables related.

Compared to patients admitted before the pandemic or in the post-vaccination phase, those hospitalised especially in the pre-vaccination phase used home respiratory support therapies more often and had higher levels of dyspnoea and other comorbidities. Perhaps these findings can be explained by the limited access to health systems and because, as already described [5], in the early months of the pandemic some AECOPD processes were handled by telephone and so only patients with more serious disease would have been hospitalised. The fact that patients admitted during the pandemic, especially in the post-vaccination phase, had fewer previous admissions might be because of the overall more general decrease in cases with hospitalisations [3–10].

Most of the cases of AECOPD had been caused by bacterial or viral infections or a combination of both [1], so the measures established in the early part of the pandemic, including restrictions on interpersonal contact, increased hand hygiene and use of masks, could have resulted in fewer infectious processes in these patients, thereby largely explaining the decrease in AECOPD [3–10]. However, to date, no work that included microbiological studies has been carried out in patients with severe AECOPD to confirm this hypothesis. Nonetheless, the largest proportion of patients with prior positive sputum cultures had been admitted in the pre-pandemic period so this could support this idea. The greater therapeutic adherence, increase in vaccination against influenza (as described in this study) and reduced industrial and urban pollution during the COVID-19 pandemic may have also been related to this finding [3–10].

The finding of higher eosinophil levels in patients admitted during the pre-vaccination phase of the pandemic may have also been related to these pre/post-pandemic differences in

**Table 4. Variables predicting Emergency Department visits or hospital readmission 3 months after discharge.**

| Variables | Total N = 333 | No Emergency Department visit / hospital readmission 90 days N = 173 | Emergency Department visit / hospital readmission 90 days N = 160 | p (no adjusted) | p (adjusted) | OR (95% IC) adjusted |
|---|---|---|---|---|---|---|
| Admission during pandemic (%) | 171 (51.4) | 79 (45.7) | 92 (57.5) | 0.03 | 0.04 | 1.6 (1.1–2.6) |
| Sex male (%) | 257 (77.2) | 136 (78.6) | 121 (75.6) | 0.44 | - | - |
| BMI > 26 (%) | 155 (47) | 99 (57.9) | 76 (47.8) | 0.08 | - | - |
| Age >71 years (%) | 173 (52) | 86 (49.7) | 87 (54.4) | 0.44 | - | - |
| Smoker/ Ex-smoker (%) | 313 (94) | 164 (98.4) | 149 (93.1) | 0.64 | - | - |
| Active smoker (%) | 121 (35.8) | 68 (41.5) | 53 (35.8) | 0.35 | - | - |
| Pack-year > 50 (%) | 160 (51.8) | 85 (52.1) | 75 (51.4) | 0.91 | - | - |
| ≥1 admission within the previous year for COPD exacerbation (%) | 116 (34.8) | 52 (30.1) | 64 (40) | 0.05 | 0.55 | 1.1 (0.7–1.9) |
| Positive sputum culture within the previous year to admission (%) | 76 (22.9) | 32 (18.5) | 44 (27.7) | 0.04 | 0.19 | 1.4 (0.8–2.6) |
| Influenza vaccination (%) | 253 (76.2) | 130 (75.1) | 123 (77.4) | 0.69 | - | - |
| Pneumococcal vaccination (%) | 227 (68.8) | 114 (66.3) | 113 (75.1) | 0.34 | - | - |
| Coexistence of some comorbidity (%) | 168 (50.6) | 79 (45.7) | 89 (56) | 0.05 | 0.21 | 1.3 (0.8–2.2) |
| FEV1 value < 43% (%) | 178 (55) | 94 (55.6) | 84 (54.5) | 0.91 | - | - |
| CAT score > 23 (%) | 175 (55.2) | 73 (43.2) | 77 (52.1) | 0.14 | - | - |
| Dyspnea mMRC 3–4 (%) | 130 (52) | 51 (29.8) | 79 (50.6) | <0.001 | | 1.9 (1.1–3.3) |
| Eosinophils > 100 cells·μL$^{-1}$ (%) | 195 (58) | 94 (54.3) | 101 (63.9) | 0.09 | - | - |
| Triple inhaled therapy at discharge (%) | 173 (52) | 106 (61.3) | 100 (62.5) | 0.82 | - | - |
| Oxygen therapy or non-invasive mechanical ventilation at discharge (%) | 176 (53) | 80 (46.2) | 96 (60.4) | 0.01 | 0.56 | 1.1 (0.7–1.9) |
| Secondary/University studies (%) | 59 (17.8) | 35 (20.4) | 24 (15.1) | 0.17 | - | - |
| Rural area of residence (%) | 162 (48.9) | 91 (53.2) | 78 (48.8) | 0.44 | - | - |
| Home ownership (%) | 217 (66.2) | 117 (68) | 100 (64.1) | 0.85 | - | - |
| Monthly income < 800€ (%) | 189 (57.4) | 88 (51.2) | 101 (64.3) | 0.02 | 0.06 | 1.6 (0.9–2.5) |
| No social relationships or only family (%) | 45 (13.5) | 17 (9.8) | 28 (17.8) | 0.05 | 0.43 | 0.7 (0.4–1.8) |
| Live alone (%) | 52 (15.6) | 24 (13.9) | 28 (17.5) | 0.37 | - | - |
| Have a caregiver (%) | 201 (60.4) | 99 (57.2) | 102 (63.8) | 0.26 | - | - |
| Informal caregiver (%)[a] | 176 (86.3) | 85 (85.9) | 91 (86.7) | >0.99 | - | - |
| Dependency for basic activities (%) | 93 (27.9) | 36 (20.8) | 57 (35.6) | 0.003 | 0.95 | 0.9 (0.5–1.8) |
| Dependency for instrumental activities (%) | 114 (71.3) | 116 (67.1) | 114 (71.3) | 0.47 | - | - |
| Previous use of social services resources (%) | 84 (25.2) | 35 (20.5) | 49 (30.6) | 0.04 | 0.52 | 1.2 (0.7–2.1) |
| Days of hospital stay ≥ 7 days (%) | 151 (45.3) | 66 (38.2) | 85 (53.1) | 0.008 | 0.14 | 1.4 (0.9–2.3) |

[a]Applied to 201 patients who had a caregiver.

behaviour. Although cases of AECOPD are typically associated with increased neutrophilic airway inflammation, eosinophilia can also be present in the sputum of some patients with an AECOPD and this correlates, at least moderately, with blood eosinophil levels [21–23]. Predominantly eosinophilic exacerbations were usually associated with a lower likelihood of purulent sputum and bacterial isolates in cultures [21,23] but also with viral–bacterial coinfections [22]. Some authors claim that some viral infections can precipitate AECOPD with higher levels of eosinophils [21], although this evidence contrasts with some classic studies [24]. Unfortunately, in our study, and with the exception of SARS-CoV-2, no systematic microbiological analyses for infections or coinfections were performed that could have allowed us to better clarify this finding.

On a social level, more patients had informal caregivers during the pandemic and this could be explained by the state of emergency, which limited the mobility of professional caregivers. The help provided by those close to these patients to reduce contact with non-cohabitants, given the vulnerability of these patients to SARS-CoV-2, also probably increased their levels of informal care [14]. Several studies have described an increased psychosocial impact and overload among informal caregivers during the pandemic, which was especially pronounced in patients that were highly dependent for the basic activities of daily living and with comorbidities [13,14], a profile that does not describe the participants in this current study.

Although we did not use specific questionnaires to measure the level of anxiety or depression of the patients in our cohorts, the number of patients diagnosed with these pathologies was similar in both periods (around 20%). These results were similar to the findings of a study conducted in Spain in non-hospitalised patients with COPD during the period of population confinement in the first wave of the COVID-19 pandemic [25]. The greater capacity to carry out the instrumental activities of daily life related to housework, as found in this current study, could perhaps also be explained by the fact that the patients spent more time at home and less care was provided by professional caregivers.

It is worth noting that the results from studies of large national databases regarding the influence of the COVID-19 pandemic on mortality caused by a severe AECOPD were contradictory [6–9,26]. In a French study, which also included patients admitted to intensive care units and cases with SARS-CoV-2 infection, hospital mortality went from 6.2% in the pre-pandemic period to 7.6% during the pandemic (relative risk: 1.24, 95% IC = 1.21–1.27) [6]. In contrast, in a Slovenian study, the non-COVID-19-related mortality of patients with COPD decreased by 15% during the pandemic [9]. Another study from Scotland and Wales found that the number of deaths from AECOPD during the pandemic was similar to that of previous years [8]. A retrospective cohort study conducted in a hospital in Malta found an increase in COPD mortality by 19.3% among 119 patients studied between 1 March and 10 May 2020 and by 8.4% compared to the 260 patients hospitalised during the same period in 2019, although the reasons for these differences were not explained by adjusting for the other variables [26]. Of note, this latter study also included patients admitted to the ICU and some with a SARS-CoV-2 coinfection [26].

In our work, in-hospital mortality caused by AECOPD was less than 2% in both periods, similar to that reported in another previous study from 2017 [16] or to that described in the same years in an analysis using big data techniques to analyse the patients admitted to the Pneumology Department in a region in the centre of Spain [27,28]. In this latter research, the in-hospital mortality of patients with COPD was 1.6% in the Pneumology Department and 8.4% in the Internal Medicine Department, with older patients and more comorbidities being more often admitted to these departments, data consistent with the results we present here [27,28]. In our work, mortality in the first 3 months after discharge was similar in the different

periods studied (≈5–7%) and was also in line with that reported in large European studies such as the COPD Audit [29].

However, it appears that patients with AECOPD admitted during the pandemic required more Emergency Department visits and readmissions than those before the pandemic, being the admission in this period an independent factor to the need for specialised care. This finding, not previously described, may be because of the poor continuity of care and follow-up of patients after hospitalisation. This may have occurred because of the difficulty of accessing other health services (primary care or day hospitals for chronic patients), an increased focus on pathologies directly related to COVID-19 and re-orientation towards remote consultation formats, making urgent hospital resources the most easily accessible route for patients in case of worsening symptoms [30].

Of note, this present study had some limitations, including that it was only conducted in two centres and that we compared two groups of patients admitted during different periods of the year. This is important because some of the differences described may have been related to the seasonality of AECOPD given that the pre-pandemic period included patients hospitalised between November 2019 and March 2020. However, some studies have shown that the pandemic also produced changes in the normal seasonality of AECOPD [9], conditioned by the general decrease in viral infections, which we believe better explains the differences we described. Variables such as pulmonary arterial hypertension, pO2 or pCO2 values were not included since not all patients had an echocardiogram or blood gas at the time of admission and at discharge, these being factors that could influence readmissions. However, indirect data related to the above, such as the need for oxygen therapy or non-invasive ventilation at discharge, did not demonstrate risk factors for readmission. The same happens with the differences in the treatments received during admission and in the previous months, although it was not shown that treatment with triple inhaled therapy was a factor associated with readmissions and all patients were treated at the discretion of their responsible physician according to current guidelines. The possible influence on readmissions of the decompensation of related comorbidities during admission, such as atrial fibrillation or heart failure, was not analysed either, although no differences were found in the history of these pathologies. No data was collected about the rehabilitation programs that could condition differences in the readmissions, although all the patients received respiratory rehabilitation during admission. Not all the patients included had a CT, so the presence of associated entities such as bullae or bronchiectasis was not analysed. The results of this study are derived from patients with COPD and, although it is possible that the social and health measures derived from the pandemic have affected patients with other respiratory diseases in a similar way, the results cannot be extrapolated to other pathologies. Nonetheless, the strength of this current work was that it prospectively collected multiple detailed clinical and, above all, social variables regarding patients who were admitted to Pneumology Departments for an AECOPD before and after the declaration of the state of healthcare emergency because of SARS-CoV-2. In addition, we also followed up these patients in the medium term to provide information that, to the best of our knowledge, is not reported elsewhere.

## Conclusion

During the COVID-19 pandemic, especially in the early months, there were some changes in the clinical and social profiles of the patients hospitalised in the Pneumology Department for AECOPD for reasons other than SARS-CoV-2, as well as their subsequent evolution. These differences may have been related to a possible decrease in infectious exacerbations and changes in the healthcare systems and in how people related to each other during this time.

Nevertheless, we found no differences in terms of poverty or loneliness. These changes were sufficient to increase subsequent hospital visits, although they did not produce an increase in mortality in the immediate period or in the following months. Nonetheless, more studies will be needed to confirm the findings described in this current work.

## Supporting information

**S1 File.**
(SAV)

## Author Contributions

**Conceptualization:** Alberto Fernández Villar, Rafael Golpe Gómez, Almudena González Montaos, Sara Fernández García, Luis Pazos Area, Ana Priegue Carrera, Alberto Ruano Raviña, Cristina Represas Represas.

**Data curation:** Alberto Fernández Villar, Rafael Golpe Gómez, Almudena González Montaos, Sara Fernández García, Luis Pazos Area, Ana Priegue Carrera, Alberto Ruano Raviña, Cristina Represas Represas.

**Formal analysis:** Alberto Fernández Villar, Rafael Golpe Gómez, Almudena González Montaos, Sara Fernández García, Luis Pazos Area, Ana Priegue Carrera, Alberto Ruano Raviña, Cristina Represas Represas.

**Funding acquisition:** Alberto Fernández Villar.

**Investigation:** Alberto Fernández Villar, Rafael Golpe Gómez, Almudena González Montaos, Sara Fernández García, Luis Pazos Area, Alberto Ruano Raviña, Cristina Represas Represas.

**Methodology:** Alberto Fernández Villar, Rafael Golpe Gómez, Almudena González Montaos, Luis Pazos Area, Ana Priegue Carrera, Cristina Represas Represas.

**Project administration:** Alberto Fernández Villar, Alberto Ruano Raviña, Cristina Represas Represas.

**Supervision:** Alberto Fernández Villar, Cristina Represas Represas.

**Validation:** Alberto Fernández Villar, Rafael Golpe Gómez, Almudena González Montaos, Sara Fernández García, Luis Pazos Area, Ana Priegue Carrera, Alberto Ruano Raviña, Cristina Represas Represas.

**Visualization:** Almudena González Montaos, Cristina Represas Represas.

**Writing – original draft:** Alberto Fernández Villar, Almudena González Montaos.

**Writing – review & editing:** Alberto Fernández Villar, Almudena González Montaos, Cristina Represas Represas.

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
