## [Decision Letter · Decision Letter 0]

4 Apr 2023

PONE-D-22-34961THE IMPACT OF THE SARS-CoV-2 PANDEMIC ON THE DEMOGRAPHIC, CLINICAL AND SOCIAL PROFILES OF PATIENTS ADMITTED TO THE PNEUMOLOGY DEPARTMENT FOR A COPD EXACERBATIONPLOS ONE

Dear Dr. González Montaos,

Thank you for submitting your manuscript to PLOS ONE. After careful consideration, we feel that it has merit but does not fully meet PLOS ONE’s publication criteria as it currently stands. Therefore, we invite you to submit a revised version of the manuscript that addresses the points raised during the review process.

We look forward to receiving your revised manuscript.

Kind regards,

Reaz Mahmud, MBBS, FCPS (Medicine), MD (Neurology)

Academic Editor

PLOS ONE

Journal Requirements:

“None of the authors declare conflicts of interest for the contents of this manuscript.

AFV has received honoraria in the last 3 years for lecturing, scientific consulting, participating in clinical studies, or writing publications for (alphabetical order): AstraZeneca, Boehringer Ingelheim, Chiesi, GlaxoSmithKline, Grifols, and Menarini.

CRR has received honoraria in the past 3 years for lecturing, scientific consulting, clinical trial participation, or publication writing for (alphabetical order): AstraZeneca, Boehringer Ingelheim, Chiesi, Faes farma, and GlaxoSmithKline.”

7. In your Data Availability statement, you have not specified where the minimal data set underlying the results described in your manuscript can be found. PLOS defines a study's minimal data set as the underlying data used to reach the conclusions drawn in the manuscript and any additional data required to replicate the reported study findings in their entirety. All PLOS journals require that the minimal data set be made fully available. For more information about our data policy, please see http://journals.plos.org/plosone/s/data-availability.

Additional Editor Comments:

Please rewrite the title according to the journal requirement

“Titles should be written in sentence case (only the first word of the text, proper nouns, and genus names are capitalized).”

The word AEPOC should be written in complete form in line 45.

In line 49, you used AECOPD, which also needs to be written in complete form.

Line 47 seems grammatically incorrect.

Line 51 started with 340; a line should not start with a number.

Line-105, please provide the ethical clearance number

Please give a patient selection flow chart in the result section.

Line 147 needs correction.

Please provide the actual results figure and p-value within the description.

Give a result summary at the beginning of the discussion.

Define some variables in the legends of the tables.

Please copy edit the manuscript for grammatical error.

Reviewers' comments:

Reviewer's Responses to Questions

**Comments to the Author**

1. Is the manuscript technically sound, and do the data support the conclusions?

Reviewer #1: Yes

Reviewer #2: Yes

Reviewer #3: Yes

Reviewer #4: Yes

2. Has the statistical analysis been performed appropriately and rigorously? 

Reviewer #1: Yes

Reviewer #2: Yes

Reviewer #3: Yes

Reviewer #4: Yes

3. Have the authors made all data underlying the findings in their manuscript fully available?

Reviewer #1: Yes

Reviewer #2: Yes

Reviewer #3: Yes

Reviewer #4: Yes

4. Is the manuscript presented in an intelligible fashion and written in standard English?

Reviewer #1: Yes

Reviewer #2: Yes

Reviewer #3: Yes

Reviewer #4: Yes

5. Review Comments to the Author

Reviewer #1: Good work. The authors have taken effort to collect data during the COVID pandemic and analysed. Statistical analysis was done appropriately. We can see all the data in the manuscript. It has been written in standard English.

Reviewer #2: This manuscript was very well done. The COVID pandemic has had vast impacts on the types of patients that are admitted to hospitals, and this paper provides insight into an important and relevant topic in today's healthcare climate.

Reviewer #3: This is a verry interesting study in which the authors have analyzed the profile of acute exacerbations in patients with COPD during the COVID-19 pandemic. The investigation has been meticulously conducted. The methodologies are sound and the study population is adequate for the main study question. The potential limitations in the study have also been discussed.

I do not really have any major comments to raise.

I would only like to ask the authors whether the potential systemic manifestations of these patients were analyzed during the study period and whether any differences could be analyzed or at least envisaged. I would like to ask about whether these results could be extrapolated to other chronic airway diseases such as asthma or bronchiectasis.

Reviewer #4: The study is good considering that we do not have many such studies on COPD and the impact that the pandemic had on it.The study was just two centric and perhaps we require studies from many more centres to arrive at some valid conclusions.The study could have been spread out over a longer time frame to avoid seasonal bias and could have included spirometry for assessing disease severity.

I have a few queries for the authors .

1) How many of the patients had PAH on echo and how many of them presented with type 2 respiratory failure at the time of admission?How many had high PaCO2 and high bicarbonate at the time of discharge because these are again independent factors for readmission.

2) Did any of the patients also have associated LV failure or atrial fibrillation or multi focal atrial tachycardia and could these be important risk factors for readmission?

3) The medication history is not mentioned in detail nor severity of the disease in terms of spirometry mentioned anywhere. Medication history in terms of high dose of inhaled steroids or even systemic steroids could have had a significant impact on the study.

4) What percentage of patients were involved in a pulmonary rehabilitation programme as that could have had a beneficial effect in preventing readmissions as also the influenza and pneumococcal vaccinations.

5) What were the various causes for AECOPD because some variables like a pneumonia, pneumothorax, structural disease as in terms of bronchiectasis , bullae could have had an impact on readmissions.

6) What was the level of self health care in both the groups and did both these groups have similar access to the care facility?

6. PLOS authors have the option to publish the peer review history of their article (what does this mean?). If published, this will include your full peer review and any attached files.

Reviewer #1: No

Reviewer #2: No

Reviewer #3: No

Reviewer #4: No

---

## [Author Response · Author response to Decision Letter 0]

17 May 2023

The response to the reviewers is included in the specific document with the same name.

---

## [Editor Report · Decision Letter 1]

6 Jun 2023

PONE-D-22-34961R1THE IMPACT OF THE SARS-CoV-2 PANDEMIC ON THE DEMOGRAPHIC, CLINICAL AND SOCIAL PROFILES OF PATIENTS ADMITTED TO THE PNEUMOLOGY DEPARTMENT FOR A COPD EXACERBATIONPLOS ONE

Dear Dr. González Montaos,

Thank you for submitting your manuscript to PLOS ONE. After careful consideration, we feel that it has merit but does not fully meet PLOS ONE’s publication criteria as it currently stands. Therefore, we invite you to submit a revised version of the manuscript that addresses the points raised during the review process.

We look forward to receiving your revised manuscript.

Kind regards,

Reaz Mahmud, MBBS, FCPS (Medicine), MD (Neurology)

Academic Editor

PLOS ONE

Journal Requirements:

Additional Editor Comments (if provided):

I noticed that you changed the title only on the title page but not in the manuscript. Please make sure to update it throughout the entire document.

To comply with regulations, please share the minimum dataset you worked on in an anonymized form. You can omit sex and residence and provide age as age groups.

When expressing P values, use 1 or 0. If it falls outside those values, express it as >0.99 or <.001. Additionally, please correct "IC 95%" to "95% CI" to follow the convention.

I noticed that all tables are in the supplementary file; I did not understand why. Please adhere to PLOS ONE guideline and include them in the manuscript file directly after the paragraph where they are first cited.

Please adjust the numbering of the figure labeled as S4 in your manuscript and place it at the beginning of the results section. Additionally, ensure that the flow chart is included at the start of the results.

Remember to attach your responses to the reviewer in the submission system. Additionally, please address the reviewer's queries briefly within the manuscript, possibly in the limitations section of the discussion or other sections as appropriate. The reviewer's questions in the manuscript were not adequately addressed and were only answered in response to the review.
---

## [Author Response · Author response to Decision Letter 1]

14 Jun 2023

Responses to reviewers also added as a supplementary file.

Journal Requirements:

Done. 

Additional Editor Comments (if provided):

I noticed that you changed the title only on the title page but not in the manuscript. Please make sure to update it throughout the entire document.

The title of the manuscript has been revised to be consistent in all the documents submitted.

To comply with regulations, please share the minimum dataset you worked on in an anonymized form. You can omit sex and residence and provide age as age groups.

The anonymized database (in Spanish) is attached as supplementary material, omitting information that allows the identification of the participants and following the indications of the ethics committee. The database is also encrypted to increase the security of the data sent. A supplementary document containing the database password is added to ensure its security.

When expressing P values, use 1 or 0. If it falls outside those values, express it as >0.99 or <.001. Additionally, please correct "IC 95%" to "95% CI" to follow the convention.

Done. 

I noticed that all tables are in the supplementary file; I did not understand why. Please adhere to PLOS ONE guideline and include them in the manuscript file directly after the paragraph where they are first cited. Please adjust the numbering of the figure labeled as S4 in your manuscript and place it at the beginning of the results section. Additionally, ensure that the flow chart is included at the start of the results.

The distribution of the tables has been changed as indicated, also adjusting their nomenclature and including the flowchart at the beginning of the results.

Remember to attach your responses to the reviewer in the submission system. Additionally, please address the reviewer's queries briefly within the manuscript, possibly in the limitations section of the discussion or other sections as appropriate. The reviewer's questions in the manuscript were not adequately addressed and were only answered in response to the review.

Issues noted by the reviewers have been added to the discussion in the limitations section.

---

## [Decision Letter · Decision Letter 2]

30 Jun 2023

PONE-D-22-34961R2The impact of the SARS-CoV-2 pandemic on the demographic, clinical and social profiles of patients admitted to the Pneumology Department for a COPD exacerbationPLOS ONE

Dear Dr. González Montaos,

Thank you for submitting your manuscript to PLOS ONE. After careful consideration, we feel that it has merit but does not fully meet PLOS ONE’s publication criteria as it currently stands. Therefore, we invite you to submit a revised version of the manuscript that addresses the points raised during the review process.

We look forward to receiving your revised manuscript.

Kind regards,

Reaz Mahmud, MBBS, FCPS (Medicine), MD (Neurology)

Academic Editor

PLOS ONE

Journal Requirements:

Additional Editor Comments:

You've assigned roles to the authors in your manuscript, but not all of them met the criteria for authorship as outlined by the International Committee of Medical Journal Editors (ICMJE) Recommendations. The four criteria for authorship include substantial contributions to the conception, design, acquisition, analysis, or interpretation of data for the work; drafting or critically revising the work for important intellectual content; final approval of the version to be published; and agreement to be accountable for all aspects of the work.

Additionally, your reference list contains 29 references, but you cited 30 in the text. Please ensure consistency between your reference list and citations.

Lastly, please submit the figure file separately and include only the figure caption in the manuscript after it has been first cited. Refer to the Plos One instructions for submission of figures for guidance.

Reviewers' comments:

Reviewer's Responses to Questions

**Comments to the Author**

1. If the authors have adequately addressed your comments raised in a previous round of review and you feel that this manuscript is now acceptable for publication, you may indicate that here to bypass the “Comments to the Author” section, enter your conflict of interest statement in the “Confidential to Editor” section, and submit your "Accept" recommendation.

Reviewer #4: All comments have been addressed

2. Is the manuscript technically sound, and do the data support the conclusions?

Reviewer #4: Yes

3. Has the statistical analysis been performed appropriately and rigorously? 

Reviewer #4: Yes

4. Have the authors made all data underlying the findings in their manuscript fully available?

Reviewer #4: Yes

5. Is the manuscript presented in an intelligible fashion and written in standard English?

Reviewer #4: Yes

6. Review Comments to the Author

Reviewer #4: All the comments have been adequately addressed by the authors and I feel the manuscript should be accepted for publication if the same is felt by the editorial board.

7. PLOS authors have the option to publish the peer review history of their article (what does this mean?). If published, this will include your full peer review and any attached files.

Reviewer #4: No

---

## [Author Response · Author response to Decision Letter 2]

28 Jul 2023

All requests for corrections have been reviewed in the manuscript. They are attached in the submitted files.

---

## [Editor Report · Decision Letter 3]

3 Aug 2023

The impact of the SARS-CoV-2 pandemic on the demographic, clinical and social profiles of patients admitted to the Pneumology Department for a COPD exacerbation

PONE-D-22-34961R3

Dear Dr. González Montaos,

We’re pleased to inform you that your manuscript has been judged scientifically suitable for publication and will be formally accepted for publication once it meets all outstanding technical requirements.

Kind regards,

Reaz Mahmud, MBBS, FCPS (Medicine), MD (Neurology)

Academic Editor

PLOS ONE
---

## [Editor Report · Acceptance letter]

7 Sep 2023

PONE-D-22-34961R3 

The impact of the SARS-CoV-2 pandemic on the demographic, clinical and social profiles of patients admitted to the Pneumology Department for a COPD exacerbation 

Dear Dr. González Montaos:

I'm pleased to inform you that your manuscript has been deemed suitable for publication in PLOS ONE. Congratulations! Your manuscript is now with our production department. 

Kind regards, 

on behalf of

Dr. Reaz Mahmud 

Academic Editor

PLOS ONE